# Policy and programmatic directions for the Lesotho tuberculosis programme: Findings of the national tuberculosis prevalence survey, 2019

R. Matji[1], L. Maama[2], G. Roscigno[3], M. Lerotholi[4], M. Agonafir[3], R. Sekibira[5], I. Law[6], M. Tadolini[7,8], N. Kak[9]*

1 AQUITY Innovations, Pretoria, South Africa, 2 Ministry of Health, Maseru, Lesotho, 3 NEXT2People, Geneva, Switzerland, 4 Scot Hospital, Maseru, Lesotho, 5 Landsat ICT Solutions, Kampala, Uganda, 6 Global Tuberculosis Programme, World Health Organization, Geneva, Switzerland, 7 Infectious Diseases Unit, IRCCS Azienda Ospedaliero-Universitaria di Bologna, Bologna, Italy, 8 Department of Medical and Surgical Sciences, Alma Mater Studiorum University of Bologna, Bologna, Italy, 9 AQUITY Global Inc., Bethesda, Maryland, United States of America

* nkak@outlook.com

**Data Availability Statement:** Data are available from Dryad (doi:10.5061/dryad.905qfttnq).

## Abstract

### Introduction

The Kingdom of Lesotho has one of the highest burdens of tuberculosis (TB) in the world. A national TB prevalence survey was conducted to estimate the prevalence of bacteriologically confirmed pulmonary TB disease among those ≥15 years of age in 2019.

### Method

A multistage cluster-based cross-sectional survey where residents ≥15 years in 54 clusters sampled from across the country were eligible to participate. Survey participants were screened using a symptom screen questionnaire and digital chest X-ray (CXR). Respondents who acknowledged cough of any duration, fever, weight loss, night sweats and/or had any CXR abnormality in the lungs were asked to provide two spot sputum specimens. All sputum testing was conducted at the National TB Reference Laboratory (NTRL), where samples underwent Xpert MTB/RIF Ultra (1st sample) and MGIT culture (2nd sample). HIV counselling and testing was offered to all survey participants. TB cases were those with *Mycobacterium tuberculosis* complex-positive samples with culture; and where culture was not positive, Xpert MTB/RIF Ultra (Xpert Ultra) was positive with a CXR suggestive of active TB and no current or prior history of TB.

### Result

A total of 39,902 individuals were enumerated, and of these, 26,857 (67.3%) were eligible to participate; 21,719 (80.9%) participated in the survey of which 8,599 (40%) were males and 13,120 (60%) were females. All 21,719 (100%) survey participants underwent symptom screening and a total of 21,344 participants (98.3%) had a CXR. Of the 7,584 (34.9%)

**Funding:** The Lesotho national tuberculosis prevalence survey was funded by the Global Fund and the World Bank Southern Africa Health System Support project (SATBHSS).

**Competing interests:** No authors have competing interests.

participants who were eligible for sputum examination, 4,190 (55.2%) were eligible by CXR only, 1,455 (19.2%) by symptom screening, 1,630 by both, and 309 by CXR exemption. A total of 6,780 (89.4%) submitted two sputum specimens, and 311 (4.1%) submitted one sample only. From the 21,719 survey participants, HIV counseling and testing was offered to 17,048, and 3,915 (23.0%) were documented as HIV-positive. The survey identified 132 participants with bacteriologically confirmed pulmonary TB thus providing an estimated prevalence of 581 per 100,000 population (95% CI 466–696) for those ≥15 years in 2019. Using the survey results, TB incidence was re-estimated to be 654 per 100,000 (95% CI 406–959), which was comparable to the 2018 TB incidence rate of 611 per 100,000 (95% CI 395–872) reported by the World Health Organization (WHO). The highest TB burden was found in those ≥55 years and among men. The ratio of prevalence to case notification was estimated at 1.22. TB/HIV coinfection was identified in 39 (29.6%) participants. Out of the 1,825 participants who reported a cough, 50% of these participants, mostly men, did not seek care. Those who sought care predominantly went to the public health facilities.

## Conclusion

The TB prevalence survey results confirmed that burden of TB and TB/HIV coinfection remains very high in Lesotho. Given that TB prevalence remains high, and there is a significant proportion of participants with confirmed TB that did not report TB suggestive symptoms. The National TB Programme will need to update its TB screening and treatment algorithms to achieve the End TB targets. A major focus will need to be placed on finding the "missing cases" i.e., undiagnosed or under-reported TB cases, or ensuring that not only TB symptomatic but also those who do not present with typical TB symptoms are promptly identified to reduce further onward transmission.

## Introduction

The Kingdom of Lesotho, with a population of 2.2 million, is a small, landlocked mountainous country encircled by South Africa. Lesotho is classified as a lower-middle-income-country with 32 percent of the population living in urban/peri-urban areas and the rest in rural areas [1]. Most of the rural population is engaged in informal crop cultivation and animal husbandry and resides in scattered small communities making delivery of health care services a challenging task [2]. Lesotho is on the list of the world's high-burden countries for both TB and HIV [3].

TB incidence increased from 827 per 100,000 (95% CI: 443–1330) in 2000 to a peak of 1,240 per 100,000 cases (95% CI: 449–2420) in 2008 [3]. This increase was largely attributed to the HIV epidemic but has steadily declined due to improved case detection, improvements in the health system's diagnostic capacity and better access to care through the decentralization of health facilities. Nonetheless the burden of TB still remains high which is fueled by a very high burden of HIV. In 2020, the prevalence of HIV among those 15 years and above was 22.7% corresponding to 324,000 people living with HIV [4].

National TB prevalence surveys help to estimate the burden of TB, particularly in countries where notification data obtained from national surveillance systems may not be accurate. Lesotho had never conducted a nationwide population-based TB prevalence survey. The country's TB burden estimates were mainly based on routine surveillance data which informed

WHO's estimates. The overall objective of the first national TB prevalence survey was to enable the National TB Programme (NTP) to gain a better understanding of the burden of TB and to identify ways of improving TB management in the country by estimating the prevalence of bacteriologically confirmed pulmonary TB disease among those ≥15 years in 2019. A secondary survey objective was to describe the healthcare seeking behaviour of survey participants. A robust knowledge of the epidemic and its determinants would help the NTP in streamlining future strategies to better address the burden of TB in Lesotho and get closer to country targets to end TB.

## Methods

### Survey design and population

The national TB prevalence survey, using multi-stage cluster sampling with probability proportional to size (PPS), was conducted between March through November 2019 based on WHO guidelines [5]. No areas of Lesotho were excluded from the sampling frame. A sample size of 26,848 individuals across 54 clusters was estimated from the following parameters: a prior guess of bacteriologically confirmed TB prevalence of 736 per 100,000 population, an average cluster size of 500, a design effect of 2.33, a relative precision on 23% and an expected participation rate of 85%. Clusters were allocated proportional to population size across 3 geographical stratum: rural (29), urban (21), peri-urban (4) (**Fig 1**). The survey population consisted of adults 15 years or more who had lived and slept in the households of selected clusters for at least two weeks prior to screening at that site.

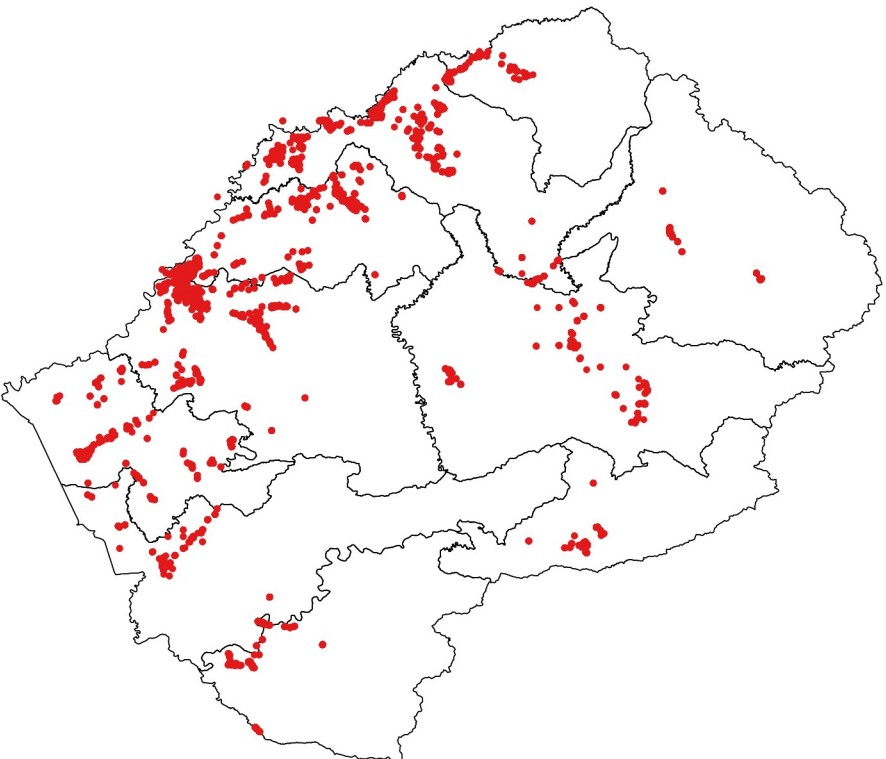

**Fig 1. Lesotho national tuberculosis prevalence survey map.** Black lines delineate provinces and red dots represent households of the enumerated survey population.

## Survey procedures

There were three survey teams and at any given time two were in the field, while the third team was preparing for its next two clusters. Each cluster was surveyed by one team over a continuous seven-day period. The first few days of operation involved door-to-door household visits, enumerating all individuals living in the selected area, and invited those who were at least 15 years of age and resident for 14 days before the survey census. At a centrally located area in the cluster, several stations were set up to conduct the following: registration; interview; digital chest X-ray (Innomed units fitted with the Samsung Detector panels and Sedecal Dragon 5kW Digital X-ray units), clinician review, sputum collection and HIV testing. All adult participants were asked to sign an informed consent form while those between 15 and 17 had to provide assent and consent from an adult or guardian. All survey participants were screened with a questionnaire (i.e., a cough of any duration, fever, unexplained weight loss in the last one month or night sweats) and a digital chest X-ray (CXR). Participants who reported any of the four symptoms and/or had CXR findings suggestive of TB were referred to the laboratory station to produce two spot sputum samples collected in sterile Falcon tubes at least 60 minutes apart. Participants who declined CXR were also asked to provide two sputum samples irrespective of whether they had TB symptoms or not. A radiologist based in South Africa read all received abnormal CXRs and approximately 11% of normal CXRs (sampled by the digital system) for quality assurance using a standardized assessment form.

## Laboratory procedures

All survey participants were offered HIV testing directly at they survey site after consenting (Alere™ Determine and UniGold™ Recombigen HIV test kits). All participants who tested HIV-positive were referred to the nearest health facility for enrollment on care, and those tested HIV-negative were counseled about HIV infection prevention.

All sputum samples were transported using a longstanding partner called "Riders for Health" (https://www.riders.org) in cool boxes to the National TB Reference Laboratory (NTRL) in Maseru. Spot 1 sample was tested with Xpert MTB/RIF Ultra (Xpert Ultra). Xpert Ultra trace results were classified as Xpert Ultra negative. Spot 2 sample was first decontaminated and processed for BD BACTEC MGIT 960 (one tube). A culture was only reported negative if there was no growth after eight weeks. For the positive cultures, identification of *Mycobacterium tuberculosis complex* was based on presumptive phenotypic appearance of colonies in the medium and confirmed using TB antigen test SD Bioline®.

## Survey TB definitions

Cases were defined in two ways: 1) All cultures with MTB-positive results (regardless of Xpert Ultra results), or 2) (a) Xpert Ultra positive (excluding trace), and (b) culture not TB (i.e., culture-negative, nontuberculous mycobacteria, contaminated, or not done), and (c) CXR suggestive of active TB as determined by three central readers, and (d) no history of TB (past or current).

## Data collection and analysis

Data were directly captured using tablets and laptops with a secure web-based application (RedCap 8.7.1). The data capture system had built in algorithms to validate data in real time as it was being captured. The central data management team cleaned and merged all data from the field, laboratory and radiology teams. Digital CXR images were stored in a cloud-based server. Data were analysed using STATA MP Version 16.1 (Stata Corp. Station TX, USA).

Best-practice analytical methods to estimate TB prevalence (i.e. number of survey participants with TB per 100,000 population) were used to account for cluster sampling, non-participation, and missing data [6]. Three logistic regression models were conducted: (1) cluster-level analysis, (2) individual-level analysis, and (3) estimation with inverse probability weighting and with multiple value imputation, and the survey population was standardized to the 2016 national population. Only model 3 is presented in this paper. As a proxy indicator of case detection, prevalence to case notification ratios (P:N) were estimated by comparing the age- and sex-specific TB prevalence to TB case notification rates for new and relapse TB cases for the same age groups and sex as reported by the NTP in 2019 [7].

### Ethical clearance and consent

The study protocol was reviewed and approved by the Lesotho Research and Ethics Committee (Reference number: ID23-2017, June 29, 2018). The data collection team followed strict ethical norms and sought informed consent or assent from each survey participant. During the enrolment, each participant was provided with information about the survey, benefits to the individual participant and the society. Opportunities were given to participants to ask questions and assured that their participation was voluntary, and that refusal will not affect any potential benefit accrued to the participant. Only participants who gave written consent were enrolled for the survey. Those participants who could not read or write, the consent form was translated in the local language. If a verbal consent was given, thumb print was taken in the presence of a witness for such participants. All participants with a positive laboratory test result (TB and HIV) were managed in accordance with national treatment guidelines.

## Results

### Survey participants

During the enumeration of 54 clusters, 39,902 individuals were recorded from 15,279 households. Of these, 26,857 (67.3%) were eligible to participate in the survey, of whom 21,719 (80.9%) participated in the survey (Fig 2). Participation was higher among women than men (84.2% vs 76.2%) and increased with age (Fig 3).

### Screening

Among the 21,719 study participants screened, 3085 (14.2%) screened positive by interview by acknowledging at least one of the screening symptoms (Table 1): 1825 (8.4%) reported having a cough, 1,423 (6.6%) had weight loss, 907 (4.2%) had night sweats and 845 (3.9%) had fever. Symptoms were reported more frequently among men than women and progressively increased with age.

A total of 21,344 (98.3%) participants were screened by CXR in the field. Of the 21,344 participants, 14,937 (70.0%) had a normal CXR, 5,820 (27.3%) had an abnormality in the lung field suggestive of TB, and 587 (2.8%) had an abnormal CXR that was not suggestive of TB. The abnormal CXR findings suggestive of TB were higher in men (35.2%) than women (21.2%) and increased with age with those ≥65 years having the highest percentage of abnormal CXRs (53.8%). There were 375 participants who were CXR exempt.

Of the 21,719 participants, 7,584 (34.9%) screened positive and were eligible to submit sputum samples: 1,455 (19.2%) by symptoms only, 4,190 (55.5%) by CXR only, 1,630 (21.5%) by both symptoms and CXR, and 309 (4.1%) by CXR exemption (and no symptoms).

Of all participants, there were a total of 285 (1.3%) who were currently on TB treatment, 1943 (8.9%) with a past history of TB treatment, and 62 (0.3%) had both.

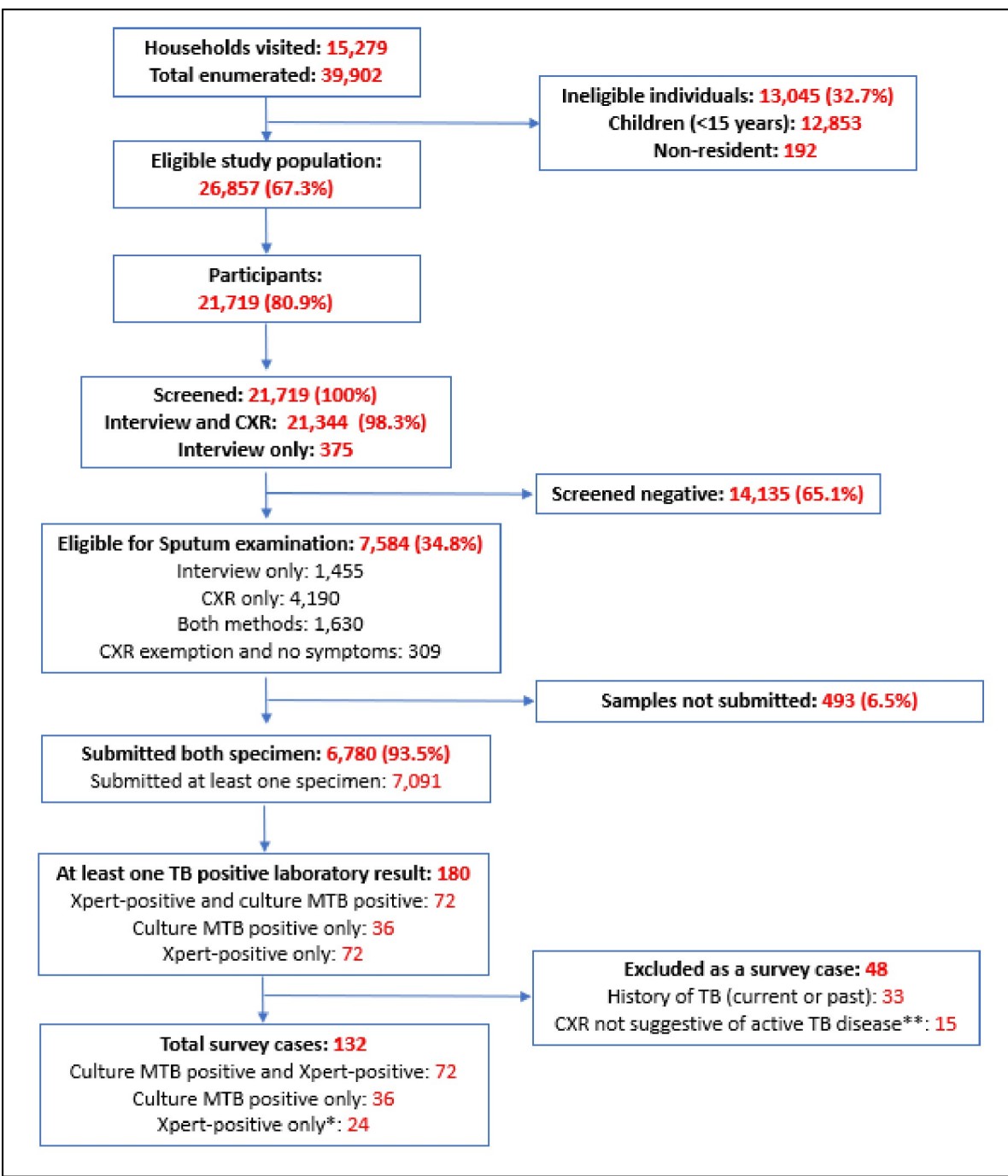

**Fig 2. Flow diagram of the national TB prevalence survey of Lesotho.** CXR, chest X-ray. *Panel review (Xpert-positive alone (no positive culture results) with a CXR suggestive of active TB disease and no current or past history of TB). ** These 15 participants also did not have a history of TB (current or past).

### Laboratory examination results

Of the 7,584 participants eligible for sputum collection, 7,091 (93.5%) submitted at least one sputum sample, 6,780 (89.4%) gave two samples, 311 (4.1%) submitted only one sample, and 493 (6.5%) did not submit any samples. Of 7,584 participants, 6,945 had a valid Xpert Ultra

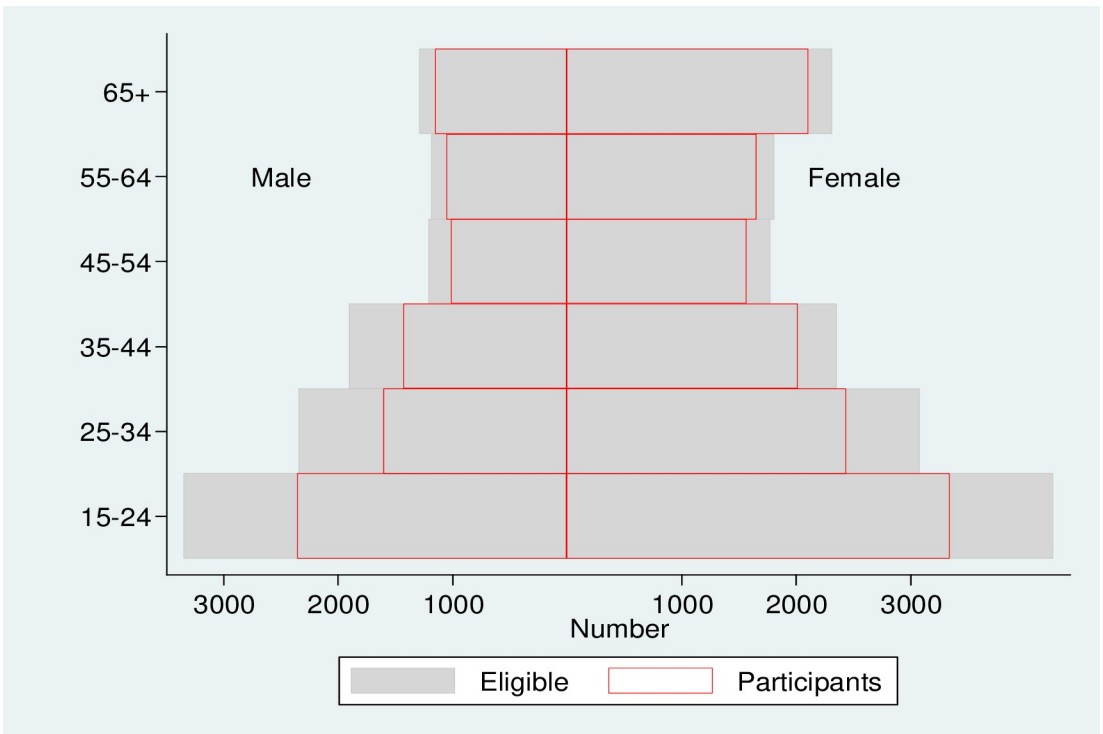

**Fig 3. Age and sex distribution of eligible and participant populations.**

result. Of these, 6,730 (96.9%) tests were negative, 144 (2.1%) were positive (grade other than trace), and 71 (1.0%) were trace positive. Of 7,584 eligible participants, 6,941 had a culture

**Table 1. Screening outcomes by sex age group and strata.**

|  | Total screened participants | Total screened positive | | Interview positive only | | CXR positive only | | Both interview and CXR positive | | CXR exempt and with no symptoms | |
|---|---|---|---|---|---|---|---|---|---|---|---|
|  | No. | No. | % | No. | % | No. | % | No. | % | No. | % |
| **Total** | 21,719 | 7,584 | 34.9 | 1,455 | 6.7 | 4,190 | 19.3 | 1,630 | 7.5 | 309 | 1.4 |
| **Sex** | | | | | | | | | | | |
| Female | 13,122 | 3,903 | 29.7 | 879 | 6.7 | 2,050 | 15.6 | 739 | 5.6 | 235 | 1.8 |
| Male | 8,597 | 3,681 | 42.8 | 576 | 6.7 | 2,140 | 24.9 | 891 | 10.4 | 74 | 0.9 |
| **Age group (years)** | | | | | | | | | | | |
| 15–24 | 5,695 | 921 | 16.2 | 305 | 5.4 | 445 | 7.8 | 84 | 1.5 | 87 | 1.5 |
| 25–34 | 4,039 | 1,029 | 25.5 | 303 | 7.5 | 512 | 12.7 | 152 | 3.8 | 62 | 1.5 |
| 35–44 | 3,436 | 1,140 | 33.2 | 270 | 7.9 | 615 | 17.9 | 221 | 6.4 | 34 | 1.0 |
| 45–54 | 2,580 | 1,065 | 41.3 | 202 | 7.8 | 608 | 23.6 | 243 | 9.4 | 12 | 0.5 |
| 55–64 | 2,710 | 1,391 | 51.3 | 185 | 6.8 | 821 | 30.3 | 366 | 13.5 | 19 | 0.7 |
| ≥65 | 3,259 | 2,038 | 62.5 | 190 | 5.8 | 1,189 | 36.5 | 564 | 17.3 | 95 | 2.9 |
| **Strata** | | | | | | | | | | | |
| Rural | 12,490 | 4,430 | 35.5 | 709 | 5.7 | 2,502 | 20.0 | 1,001 | 8.0 | 218 | 1.7 |
| Urban | 8,230 | 2,613 | 31.7 | 634 | 7.7 | 1,439 | 17.5 | 462 | 5.6 | 78 | 0.9 |
| Peri-urban | 999 | 541 | 54.2 | 112 | 11.2 | 249 | 24.9 | 167 | 16.7 | 13 | 1.3 |

CXR, chest X-ray.

result: 6,305 (90.9%) tests were culture negative, 108 (1.6%) were culture positive for *Mycobacterium tuberculosis*, 340 (4.9%) were contaminated, and 188 (2.7%) were nontuberculous mycobacteria. Of the 180 laboratory positive results, 72 (40%) were detected by both methods, 72 (40%) were positive by Xpert Ultra alone, while 36 (20%) were culture positive for *M.tb* but negative or trace by Xpert Ultra. Assuming culture to be the reference standard, 66 out of 144 (46%) Xpert Ultra-positives (excluding trace) were culture-negative (**Table 2**). When comparing Xpert Ultra with culture, culture positivity increased with Xpert Ultra grade, and most culture positives had a low grade Xpert Ultra result.

The sensitivity and specificity of symptom screening, assuming culture was the reference standard, was 35.2% (95% CI: 26.2–45.9) and 59.9% (95% CI: 58.7–61.1%), respectively. The sensitivity and specificity of CXR screening, assuming culture was the reference standard, was 93.5% (95% CI: 87.1–97.4) and 21.2% (95% CI: 20.2–22.2), respectively.

Assuming culture was the reference standard (and trace was negative), the specificity of Xpert Ultra among those with no history of TB was 99.3% (95% CI: 99.1–99.5). This was lower among those with a history of TB (current or past history of TB): 97.3% (95% CI 96.1–98.0).

## Participants identified with TB

There were a total of 180 participants with either a positive result by culture or Xpert Ultra. There were 108 culture TB positive (of which 72 were also positive by Xpert Ultra). Of the remaining 72 participants who were Xpert Ultra-positive alone, 33 were excluded as cases because they had a current or past history of TB, and the other 39 CXRs were re-examined by a three-person panel of which 24 were found to have CXRs suggestive of active TB disease. Therefore, a total of 132 participants had bacteriologically confirmed pulmonary TB based on the survey case definition: 36 (27.3%) by culture alone, 72 (54.6%) were positive by Xpert Ultra and culture; and 24 (18.2%) by Xpert Ultra alone (no positive culture results) with a CXR suggestive of active TB disease and no current or past history of TB. A total of 6 survey participants were identified via Xpert Ultra as rifampicin resistant.

Yield was highest among those detected via CXR screening which identified 125 participants with TB (94.7%), followed by symptom screening which identified 49 others (37.1%) (**Table 3**). CXR alone identified 78 participants with TB, symptoms alone identified 2, and 5 (3.8%) others were CXR exempt and reported no symptoms. Of the 49 symptomatic

**Table 2. Xpert MTB/RIF Ultra and MGIT culture results among those who screened positive.**

| Xpert MTB/RIF Ultra results | Culture results | | | | | | |
|---|---|---|---|---|---|---|---|
| | Culture-positive for *Mycobacterium tuberculosis* | Culture-negative for *Mycobacterium tuberculosis* | Non-tuberculous mycobacteria | Contaminated | Not done | Rejected | Total |
| Xpert MTB/RIF Ultra positive | 72 | 66 | 0 | 6 | 0 | 0 | **144** |
| High | 10 | 2 | 0 | 0 | 0 | 0 | **12** |
| Medium | 11 | 6 | 0 | 0 | 0 | 0 | **17** |
| Low | 36 | 29 | 0 | 3 | 0 | 0 | **68** |
| Very low | 15 | 29 | 0 | 3 | 0 | 0 | 47 |
| Xpert MTB/RIF Ultra negative | 22 | 6,182 | 187 | 332 | 6 | 1 | **6,730** |
| Trace* | 14 | 54 | 1 | 2 | 0 | 0 | **71** |
| Not done | 0 | 3 | 0 | 0 | 635 | 1 | **639** |
| **Total** | **108** | **6,305** | **188** | **340** | **641** | **2** | **7,584** |

*Xpert MTB/RIF Ultra trace results were classified as negative.

**Table 3. Yield by screening category and HIV status.**

| Screening category | Screen positive participants | Participants with TB | | | | | | | |
|---|---|---|---|---|---|---|---|---|---|
| | | Total | % | HIV-positive* | % | HIV-negative | % | HIV-unknown | % |
| Symptoms only | 1,455 | 2 | 0.14 | 0 | 0 | 2 | 100 | 0 | 0 |
| Abnormal CXR only | 4,190 | 78 | 1.9 | 23 | 29 | 38 | 49 | 17 | 22 |
| Symptoms and abnormal CXR | 1,630 | 47 | 2.9 | 15 | 32 | 27 | 57 | 5 | 11 |
| CXR exempt with no symptoms | 309 | 5 | 1.6 | 1 | 20 | 4 | 80 | 0 | 0 |
| Total | 7,584 | 132 | 1.7 | 39 | 30 | 71 | 54 | 22 | 17 |

*HIV status is a combination of self-reported and HIV testing.

participants with TB, 42 (86%) reported a cough, 24 (49%) reported weight loss, 16 (33%) reported fever and 15 (31%) reported night sweats.

## HIV status

A total of 16,092 (74%) of the participants knew and were willing to inform their HIV status, and of those, 3862 (18%) self-reported being HIV-positive. Complete HIV testing data were not available from the first 10 clusters as testing was initially conducted by an independent NGO that did not align with survey requirements. The survey team took over direct HIV testing thereafter. Of 17,031 participants tested for HIV in the survey, 3,915 (23.0%) were positive (of these 3,505 self-reported as positive). In total, HIV status was known for 17,915 participants: 4,048 (22.6%) were HIV-positive (of these 133 self-reported only and 3,915 tested in the survey). Of the 110 survey participants with known HIV status, 39 (29.5%) were HIV-positive and 71 (64.5%) were HIV-negative. More women than men were co-infected (17 participants, 48.6% vs. 22 participants, 31.9%), and highest in the 25–44-year age groups (20 participants, 66.7%). The proportion of symptomatic survey participants by HIV status was similar (coinfected: 38.5%, 15/39; HIV-negative: 40.8%, 29/71) (**Table 3**). Of the 39 co-infected people, 30 already knew they were HIV-positive prior to the survey and were not currently being treated for TB.

## Estimated prevalence of bacteriologically confirmed pulmonary TB

The weighted and adjusted prevalence of bacteriologically confirmed pulmonary TB among those≥15 years was 581 per 100,000 (95% CI: 466–696) (**Table 4**). Estimated prevalence was significantly higher in men than women and increased with age with notable peaks in the 35–44 and ≥65 years age groups. Prevalence was 2.5 times higher in those co-infected with HIV, compared with those who were HIV-negative. There was no difference in prevalence between geographical strata. The prevalence-to-notification (P:N) ratio was 1.22 overall, greater than 1.5 in those ≥55 years age groups, and higher for men than for women (**Table 4**).

## Health-seeking behaviour

Of 1,825 participants who reported a cough, 916 (50.2%) did not seek care for it; men (544, 59.4%), those living in rural areas (502, 54.8%), and those ≥65 years (180, 19.7%) were least likely to seek care. The most common reasons given for not seeking care were ignorance (30%), not recognized as illness (24%), and self-treatment (27%); of note, high costs only accounted for 8.2% of participants. More people with HIV and cough sought care than those without HIV and cough (54.9% vs 47.3%). Survey participants who did seek care for their symptoms primarily went to public health facilities (86%), followed by pharmacies especially in urban areas (5.8%), nongovernmental organizations (4.4%) then private health facilities (3.7%).

**Table 4. Estimated prevalence of bacteriologically confirmed pulmonary TB among individuals aged ≥15 years, and prevalence to case notification ratios.**

| Category | Number of survey participants with TB | Sex ratio (Male: Female) | Estimated prevalence, cases per 100,000 population (95% CI) | TB case notification rate per 100,000 in 2018, new and relapsed cases | Prevalence to case notification ratio |
|---|---|---|---|---|---|
| **All** | 132 | 2.0 | 581 (466–696) | 476 | 1.22 |
| **Sex** | | | | | |
| Female | 44 | n/a | 327 (218–435) | 331 | 0.99 |
| Male | 88 | n/a | 849 (642–1,057) | 628 | 1.35 |
| **Age** | | | | | |
| 15–24 | 7 | 2.5 | 142 (26–258) | 127 | 1.11 |
| 25–34 | 12 | 1.4 | 331 (147–516) | 425 | 0.78 |
| 35–44 | 25 | 1.8 | 892 (516–1,268) | 619 | 1.44 |
| 45–54 | 13 | 3.3 | 593 (267–920) | 664 | 0.89 |
| 55–64 | 28 | 3.0 | 1,211 (795–1,626) | 808 | 1.50 |
| ≥65 | 47 | 1.6 | 1,661 (1,143–2,178) | 1,061 | 1.57 |
| **Strata** | | | | | |
| Rural | 85 | 2.1 | 670 (491–848) | - | - |
| Urban | 40 | 1.7 | 453 (310–595) | - | - |
| Peri-urban | 7 | 2.5 | 531 (329–732) | - | - |
| **HIV status*** | | | | - | - |
| HIV-positive | 39 | 1.3 | 1,272 (870–1,673) | - | - |
| HIV-negative | 65 | 2.4 | 504 (364–643) | - | - |

* HIV status was not known in 28 participants with TB.

Of the 49 participants with TB who had a cough, 19 (39%) did not seek care. Of these 19, 9 (47%) self-treated, 9 (47%) did not recognize it as an illness or ignored it. Of the 30 participants with TB who had a cough and had sought care, 26 (87%) visited a public health facility (and only 1 of the 26 were on current TB treatment at the time of the survey), and pharmacies and private practitioners accounted for the remainder.

## Discussion

Lesotho is in WHO's global list of high-burden countries for TB and HIV-associated TB, and the prevalence survey further confirms this [3]. The estimated prevalence of bacteriologically confirmed pulmonary TB in those aged ≥15 years, 581 per 100,000 population (95% CI: 466–696), is comparable to recent surveys conducted in neighbouring countries with also high co-infection burdens: Eswatini (352 per 100,000 population 95% CI: 264–440), Mozambique (334, 252–416) South Africa (852, 679–1026), and Zambia (638, 502–774) [8–10]. Based on these survey data, post-survey incidence estimates were revised to 654 per 100 000 (95% CI: 406–959) by WHO; making it one of the highest in the world [11]. It is comparable to the pre-survey estimate of 2018: 611 per 100,000 (95% CI 395–872) [12].

Prevalence increased with age with a notable peak of more than 1% in those greater than 55 years of age, and a secondary peak in the working age group of 35–44-year-olds. This bimodal distribution suggests that TB in Lesotho is marked by active transmission and suggests an epidemic driven by both HIV in the younger age groups and reactivation in older ones. Over the past decade, Southern Africa have seen large programmatic investments in HIV with concomitant declines in TB incidence. Between 2010 and 2017, Lesotho had an average annual rate of decline of 7%, with notifications falling at a similar rate during that time period (3). Latest estimates from the 2020 population-based HIV impact assessment estimates that 90.1% of adults

living with HIV knew their status, 96.9% of adults who were aware of their HIV status who were on ART, and 91.5% of adults who were on ART had viral load suppression (4). Nonetheless, despite these great gains, the burden of TB/HIV still remains incredibly high. It is of note that there were 30 co-infected people who were not diagnosed with TB prior to the survey; assuming they were in HIV care, this presents missed opportunities for TB screening and diagnosis.

Needless to say, that the proportion of people with TB who are not infected with HIV is also high. Given the high ART coverage, PLHIV likely had more opportunities to access and engage with the health system, and therefore were more likely to have any TB suggestive symptoms detected and investigated, as evidenced by the higher TB/HIV co-infection rate reported by the NTP in 2019 (62%) compared to the survey findings (30%) (3). Furthermore, more PLHIV in the survey sought care for their symptoms than those without HIV, therefore, greater effort is also required to identify those with TB who are HIV-negative.

Like many other prevalence surveys, we also observed significant gender disparity in TB burden and in health-seeking behaviour related to TB care, showing greater reluctance among men to seek care when sick [13]. Among the survey's confirmed participants with TB, the majority of those with symptoms who did not seek treatment were men. Combined with the findings that men had a burden more than twice that of females (especially in young and older men) and a high ratio of prevalence to notification in men, it highlights that Lesotho needs to identify social and structural determinants (e.g. mining, smoking, alcohol, undernutrition) that will assist in developing specific approaches to remove barriers and stigmas to case finding, increase equitable access to care, reduce delays in diagnosis, and improve the retention to effective management of TB among men.

Overall, the survey identified a gap between TB prevalence and cases routinely notified to the NTP be it through underdiagnosis or underreporting. Notably, men had a lower proportion of being detected and reported with TB than women. Other gaps include those in the 35-to-44-year age group, and those aged above 55 years. The national programme will need to develop targeted strategies to reach men in general, but to also identify where people in these age groups are generally seeking care. Although we highlight the limitations and barriers of certain subgroups seeking care, another critical factor is the level of service quality especially in public health sectors. The survey identified more than 80% of symptomatic participants diagnosed with TB that had visited a public health facility. An examination of the cascade of care is required to identify why most of these people, especially men, were not diagnosed [14].

The survey also showed that a significant proportion of TB participants did not report classical TB symptoms. Most were identified due to an abnormal screening CXR (83/132, 63%), therefore a large number of TB prevalent cases would have missed detection by using only the 4-symptom screen. Considering that Lesotho uses symptom screening for identification of those with presumptive TB, this survey highlights that excluding CXR screening potentially misses a large proportion of TB cases, reinforcing the urgent need for the local adaptation of recent recommendations by WHO for routine CXR use as a sensitive TB screening tool in active case-finding [15]. This issue is similar to other TB prevalence surveys where a substantial proportion of identified people with TB are subclinical i.e., bacteriologically confirmed for TB but do not report screening symptoms, and potentially driving transmission [16].

One of the survey limitations relates to the performance characteristics of Xpert Ultra when used for active case finding. As shown in various reviews, assuming culture is the reference standard in this survey, specificity of Xpert Ultra was lower in those with a history of TB than those without [17, 18]. Therefore, the survey case definition was quite conservative in restricting cases to only those with enough evidence to minimize the test's effect i.e., Xpert Ultra-positive with no culture confirmation, no history of TB and a CXR suggestive of active TB disease.

Given the historical cumulative burden of TB in the country (and others in the region), this will continue to affect the use of Xpert Ultra in active case finding projects. This does not assume culture was perfect either. Given the challenges of geography, transportation of specimens under cold chain may impact culture performance especially given the paucibacillary nature of specimens collected via active case finding. The contamination rate was only 4.9% suggesting potential harsh decontamination of specimens. Therefore, it may be possible that some Xpert Ultra-positive only results could have had a confirmatory culture e.g., those with a high or medium Xpert Ultra grade. Despite the diagnostic challenges, estimated prevalence, albeit based on a conservative case definition, was still inextricably high.

The other major limitation was not knowing the HIV status for all survey participants from the first 10 clusters. HIV results were not available at the time of analysis because initially we relied on a Government partner to provide HIV results, but that strategy did not work. Assuming that the proportion of PLHIV were the same for all clusters, estimates of TB/HIV prevalence were conservative and still very high.

A repeat TB prevalence survey (implemented after 8–10 years) could be an excellent source of information to show gaps in disease detection and burden trends over time. However, prevalence surveys provide sufficient data to allow estimates of the burden at the national or provincial levels, but they do not provide enough data for decision making at districts or levels below. With this and other limitations in mind, including the huge cost of a single TB prevalence survey, it is critical to strengthen the national TB surveillance system and use different approaches to identify gaps in disease detection and reporting (e.g. inventory studies, capture-recapture studies, etc.) [19].

## Conclusions

The Lesotho TB prevalence survey is one of the first where both Xpert Ultra and culture were used in parallel to establish TB prevalence at the national level. It is clear that Lesotho continues to have a very high burden of TB with many undetected cases in the community, and that HIV remains a major driver of the TB epidemic. The national TB programme will need to work closely with HIV programme in order to optimize screening and diagnostic opportunities; develop strategies to increase healthcare seeking especially among those with symptoms; and undertake greater utilization of chest X-ray facilities (with concomitant training of more in-country radiologists) for active case finding activities. The other key challenges relate to gender disparity, community knowledge of TB, and access to quality diagnostics and human resources for faster and more effective case finding, patient-centered treatment and support.

## Acknowledgments

The paper was written in memory of Dr Patrick Hazangwe, medical officer of WHO South Africa, who provided technical support to the survey team. The survey team is thankful to the field data collection teams as well as participants who willingly joined this survey. We are also thankful to Aruni Liyanage and Yangchen Dolkar from URC who provided financial and administrative support for the survey implementation.

**Disclaimer:** The authors alone are responsible for the views expressed in this publication, and they do not necessarily represent the decisions or policies of their organizations.

## Author Contributions

**Conceptualization:** R. Matji, L. Maama, G. Roscigno, I. Law, M. Tadolini, N. Kak.

**Data curation:** M. Lerotholi, M. Agonafir, R. Sekibira.

**Formal analysis:** L. Maama, M. Agonafir, R. Sekibira, I. Law, N. Kak.

**Investigation:** R. Matji, M. Agonafir.

**Methodology:** R. Matji, G. Roscigno, I. Law, M. Tadolini, N. Kak.

**Project administration:** R. Matji, M. Lerotholi, N. Kak.

**Software:** R. Sekibira.

**Supervision:** R. Matji, L. Maama, M. Lerotholi, R. Sekibira.

**Writing – original draft:** R. Matji, L. Maama, G. Roscigno, I. Law, M. Tadolini, N. Kak.

**Writing – review & editing:** R. Matji, M. Lerotholi, M. Agonafir, R. Sekibira, I. Law, M. Tadolini, N. Kak.

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
