## [Decision Letter · Decision Letter 0]

28 Feb 2022

PONE-D-22-00269Policy and programmatic directions for the Lesotho tuberculosis programme: findings of the national tuberculosis prevalence survey, 2019PLOS ONE

Dear Dr. KAK,

Thank you for submitting your manuscript to PLOS ONE. After careful consideration, we feel that it has merit but does not fully meet PLOS ONE’s publication criteria as it currently stands. Therefore, we invite you to submit a revised version of the manuscript that addresses the points raised during the review process.

Thank you for submitting this informative manuscript. Beyond several specific points for clarification and improvement raised by the reviewers, there appear to be some technical problems with your references, tables, and figures. Please ensure these are resolved, by carefully checking the PDFs generated from your files, before finalizing your resubmission. Please submit your revised manuscript by Apr 09 2022 11:59PM. If you will need more time than this to complete your revisions, please reply to this message or contact the journal office at plosone@plos.org. Please include the following items when submitting your revised manuscript:A rebuttal letter that responds to each point raised by the academic editor and reviewer(s). You should upload this letter as a separate file labeled 'Response to Reviewers'.A marked-up copy of your manuscript that highlights changes made to the original version. You should upload this as a separate file labeled 'Revised Manuscript with Track Changes'.An unmarked version of your revised paper without tracked changes. You should upload this as a separate file labeled 'Manuscript'.

We look forward to receiving your revised manuscript.

Kind regards,

Kevin Schwartzman

Academic Editor

PLOS ONE

Journal Requirements:

e) Please provide an amended Funding Statement that declares *all* the funding or sources of support received during this specific study (whether external or internal to your organization) as detailed online in our guide for authors at http://journals.plos.org/plosone/s/submit-now.  

(No authors have competing interests)

Please state what role the funders took in the study.  If any authors received a salary from any of your funders, please state which authors and which funder. If the funders had no role, please state: "The funders had no role in study design, data collection and analysis, decision to publish, or preparation of the manuscript." 

Please send your amended statements by return email; we will change the online submission form on your behalf. 

6. We note that Figure 1 in your submission contain [map/satellite] images which may be copyrighted. All PLOS content is published under the Creative Commons Attribution License (CC BY 4.0), which means that the manuscript, images, and Supporting Information files will be freely available online, and any third party is permitted to access, download, copy, distribute, and use these materials in any way, even commercially, with proper attribution. For these reasons, we cannot publish previously copyrighted maps or satellite images created using proprietary data, such as Google software (Google Maps, Street View, and Earth). For more information, see our copyright guidelines: http://journals.plos.org/plosone/s/licenses-and-copyright.

7. Please include your tables as part of your main manuscript and remove the individual files. Please note that supplementary tables (should remain/ be uploaded) as separate "supporting information" files

Reviewers' comments:

Reviewer's Responses to Questions

**Comments to the Author**

1. Is the manuscript technically sound, and do the data support the conclusions?

Reviewer #1: No

Reviewer #2: Yes

2. Has the statistical analysis been performed appropriately and rigorously? 

Reviewer #1: Yes

Reviewer #2: Yes

3. Have the authors made all data underlying the findings in their manuscript fully available?

Reviewer #1: Yes

Reviewer #2: Yes

4. Is the manuscript presented in an intelligible fashion and written in standard English?

Reviewer #1: No

Reviewer #2: Yes

5. Review Comments to the Author

Reviewer #1: The manuscript entitled “Policy and programmatic directions for the Lesotho tuberculosis programme: findings of the national tuberculosis prevalence survey, 2019” is an interesting study. The TB prevalence survey is important to inform estimates of TB disease burden in the 30 high TB burden countries, of which Lesotho is part. The survey was accomplished in 2019 and the data were available, according to the Global Tuberculosis Report 2021. Accompanying the report, this scientific manuscript will help global readers to learn and understand the methods used for the survey, and to obtain a clear picture of TB burden in Lesotho, like Vietnam’s publication (2020) : https://journals.plos.org/plosone/article?id=10.1371/journal.pone.0232142 .

However, after looking through the manuscript, there are many issues that authors need to address before it ready for a publication. In this stage, the manuscript is not ready for publication.

1. The introduction needs to clarify the importance of the study. Readers with adequate tuberculosis background may understand what TB prevalence survey is. But it is also important to provide sufficient information why TB prevalence survey is required in Lesotho, what is the link with the TB estimation numbers, and what is the use of the data.

2. The tables and figures used templates provided by the WHO. However, authors did not check them before submission, and now they are very confusing to be read and reviewed. It is better to first check all text, figures, and tables are formatted following the guidelines. Also please check that Table 2 and 3 are duplicated with different formats.

3. Authors need to be careful to use terms ‘prevalence’, ‘prevalence per 100 000’, and ‘prevalence rate’. They will be very different in interpretation between each other. Please check again throughout the manuscript.

4. In Discussion, it will be better to highlight that the findings from this TB prevalence survey refine the previous prevalence estimation published in the WHO Report. Also, better to discuss how it could be replicated, and should be implemented in regular period (how if ‘yes’, and why if ‘not’). Are the resources available for repeated prevalence, and if not, how to deal with it?

5. The discussion lacks references. There are many prevalence surveys published, mostly in PloS, but not cited in this paper, for example:

a. Zambia: https://journals.plos.org/plosone/article?id=10.1371/journal.pone.0146392

b. Vietnam: https://journals.plos.org/plosone/article?id=10.1371/journal.pone.0232142

c. Kenya: https://journals.plos.org/plosone/article?id=10.1371/journal.pone.0209098

d. Bangladesh: https://journals.plos.org/plosone/article?id=10.1371/journal.pone.0044980

6. References should conform with journal guidance. Please see the author guidance. There is no time pressure for refining all formatting before submission, and it will also help reviewers to provide the best comments for improvement.

7. This paper still needs English proof-editing before further submission.

Reviewer #2: Congratulations for this manuscript that reports the results of the first ever national tuberculosis (TB) prevalence survey in Lesotho. You have followed the international guidance to carry out the survey, revealed a very high estimated prevalence and found out about community health seeking behavior. The manuscript is well and clearly written and its flow is good. There a few typos that you will, no doubt, fix in due course (please check separator commas in line 270, Figure 2 and Table 3, consider writing MTB in full in Tables 2 and 3.

A couple of other points for your consideration:

MAJOR:

* You touch on HIV related issues on lines 83-84, 286 and 335 but you do not explain anywhere how Lesotho is performing on 90:90:90/95:95:95. This information is essential in view of your finding of 2,5 fold prevalence of TB among PLHIV compared with HIV-uninfected participants. Is the 'problem' the proportion of PLHIV who know their status? There must be an error when you mention of a high proportion of people 15-49 years knowing their status - 23% is low. What is the reported level of viral suppression among PLHIV In HIV care/ART? Are there issues of poor retention? In line 337, you state that the TB-HIV burden remains high - is this in line with other countries in southern Africa, the world's HIV epicenter? What does literature tell us? What is your hypothesis regarding this high burden?

* line 207: 62 participants reported both a current and past episode of TB? How does this compare with findings reported by other prevalence surveys? Is it high (in keeping with TB-HIV burden) or low? You may wish to discuss this point further.

MINOR:

* line 194: word 'than' missing?

* Conclusion: could you make 1-2 specific recommendations directed to the NTP (and general health services) about quality of health and particularly TB case finding and holding services? How can Lesotho narrow the case finding gap? What are your recommendations to policy makers?

6. PLOS authors have the option to publish the peer review history of their article (what does this mean?). If published, this will include your full peer review and any attached files.

Reviewer #1: No

Reviewer #2: No

---

## [Author Response · Author response to Decision Letter 0]

1 Jun 2022

11 May 2022

Dear Editor, 

We would like to thank you and the reviewers for the comprehensive feedback provided and for the opportunity to submit a revised version of our manuscript. Here below the point-by-point answer to the specific comments raised are in blue. 

Reviewer #1: The manuscript entitled “Policy and programmatic directions for the Lesotho tuberculosis programme: findings of the national tuberculosis prevalence survey, 2019” is an interesting study. The TB prevalence survey is important to inform estimates of TB disease burden in the 30 high TB burden countries, of which Lesotho is part. The survey was accomplished in 2019 and the data were available, according to the Global Tuberculosis Report 2021. Accompanying the report, this scientific manuscript will help global readers to learn and understand the methods used for the survey, and to obtain a clear picture of TB burden in Lesotho, like Vietnam’s publication (2020) : https://journals.plos.org/plosone/article?id=10.1371/journal.pone.0232142 .

However, after looking through the manuscript, there are many issues that authors need to address before it ready for a publication. In this stage, the manuscript is not ready for publication.

1. The introduction needs to clarify the importance of the study. Readers with adequate tuberculosis background may understand what TB prevalence survey is. But it is also important to provide sufficient information why TB prevalence survey is required in Lesotho, what is the link with the TB estimation numbers, and what is the use of the data.

Thank you. We have added additional text in the introduction to provide further explanation on why the prevalence survey was needed in Lesotho and what was the intended use of the survey results. 

2. The tables and figures used templates provided by the WHO. However, authors did not check them before submission, and now they are very confusing to be read and reviewed. It is better to first check all text, figures, and tables are formatted following the guidelines. Also please check that Table 2 and 3 are duplicated with different formats.

Thank you. The tables and figures have been inserted into the manuscript. For simplicity, tables 2 and 3 are now merged into one table, and so have tables 5 and 6. So there are now only 4 tables. Titles for table 1, 2, 3 and 4, and figure 1 have also been edited. 

3. Authors need to be careful to use terms ‘prevalence’, ‘prevalence per 100 000’, and ‘prevalence rate’. They will be very different in interpretation between each other. Please check again throughout the manuscript.

The use of prevalence in the manuscript refers to the number of survey cases per 100,000 population. This has now been defined in the methods, and explicitly noted in one of the headings of the updated table 4. Please see line 170 (clean version).

4. In Discussion, it will be better to highlight that the findings from this TB prevalence survey refine the previous prevalence estimation published in the WHO Report. Also, better to discuss how it could be replicated, and should be implemented in regular period (how if ‘yes’, and why if ‘not’). Are the resources available for repeated prevalence, and if not, how to deal with it?

Further text has been added to the discussion. Please see lines 445–451.

5. The discussion lacks references. There are many prevalence surveys published, mostly in PloS, but not cited in this paper, for example:

a. Zambia: https://journals.plos.org/plosone/article?id=10.1371/journal.pone.0146392

b. Vietnam: https://journals.plos.org/plosone/article?id=10.1371/journal.pone.0232142

c. Kenya: https://journals.plos.org/plosone/article?id=10.1371/journal.pone.0209098

d. Bangladesh: https://journals.plos.org/plosone/article?id=10.1371/journal.pone.0044980

Although we are aware of these other papers, we only wanted to draw on the comparison of surveys with a similar context i.e. Southern Africa, with a high TB/HIV burden. Hence we only referenced recent surveys from South Africa, Mozambique and Eswatini. We will also mention Zambia.

6. References should conform with journal guidance. Please see the author guidance. There is no time pressure for refining all formatting before submission, and it will also help reviewers to provide the best comments for improvement.

The reference list has now been updated to Vancouver style.

7. This paper still needs English proof-editing before further submission.

Thank you. The text has been further edited including the rephrasing of the word “cases” where appropriate.

Reviewer #2: Congratulations for this manuscript that reports the results of the first ever national tuberculosis (TB) prevalence survey in Lesotho. You have followed the international guidance to carry out the survey, revealed a very high estimated prevalence and found out about community health seeking behavior. The manuscript is well and clearly written and its flow is good. There a few typos that you will, no doubt, fix in due course (please check separator commas in line 270, Figure 2 and Table 3, consider writing MTB in full in Tables 2 and 3.

Thank you. MTB has been written out in full and thousand separators have been inserted in figures and tables, and checked throughout the paper.

A couple of other points for your consideration:

MAJOR:

* You touch on HIV related issues on lines 83-84, 286 and 335 but you do not explain anywhere how Lesotho is performing on 90:90:90/95:95:95. This information is essential in view of your finding of 2,5 fold prevalence of TB among PLHIV compared with HIV-uninfected participants. Is the 'problem' the proportion of PLHIV who know their status? There must be an error when you mention of a high proportion of people 15-49 years knowing their status - 23% is low. What is the reported level of viral suppression among PLHIV In HIV care/ART? Are there issues of poor retention? In line 337, you state that the TB-HIV burden remains high - is this in line with other countries in southern Africa, the world's HIV epicenter? What does literature tell us? What is your hypothesis regarding this high burden?

Thank you. We have inserted further text into the discussion related to the impact of HIV in Lesotho including the latest data from the 2020 population-based HIV impact assessment for Lesotho. 

Apologies. The “23%” referred to the prevalence of HIV in the general population as a whole, and has now been removed.

We already know from WHO estimates that the burden of TB/HIV in Lesotho is one of the highest in the world, and the survey confirms this. We have already mentioned in the first paragraph of the discussion that these estimates are consistent with neighbouring countries with similarly high co-infection rates.

Approximately a third of people diagnosed with TB in the survey who knew their status were HIV-positive. This is a missed opportunity for diagnosis within the HIV service, and just one of many possibilities for why coinfection burden is so high. 

See lines 375-381.

* line 207: 62 participants reported both a current and past episode of TB? How does this compare with findings reported by other prevalence surveys? Is it high (in keeping with TB-HIV burden) or low? You may wish to discuss this point further.

Thank you. The proportion of the survey population (and TB survey cases) with a current history of TB is important for the estimation of incidence. Its relevance is to quantify the chronicity of TB in the country, and the increasing number of people who have survived TB. We do not believe a comparison of these proportions warrants comparisons as the better indicator for defining burden of disease are annual notifications and estimated incidence. However, for your interest, the proportion of participants with TB that also have a current history of TB in South Africa, Eswatini and Mozambique are 4.3%, 6.1% and 7.9% respectively. In Lesotho, this proportion was 3.0 %. 

MINOR:

* line 194: word 'than' missing?

Thank you. This has been edited.

* Conclusion: could you make 1-2 specific recommendations directed to the NTP (and general health services) about quality of health and particularly TB case finding and holding services? How can Lesotho narrow the case finding gap? What are your recommendations to policy makers?

Thank you. Some additional comments have been made in the conclusion.

 

JOURNAL REQUIREMENTS

A. Responses to general comments 

1. Manuscript meets the PLOS ONE’s style requirement

We have reviewed the manuscript and believe that the revised manuscript it meets PLOS ONE’s style requirements. e.g. updating of language for clarity and references to match the Vancouver style.

2. Please include a complete copy of PLOS’s questionnaire on inclusivity on global research.

The completed questionnaire on inclusivity in global research is now attached, as requested.

3. Contradictory funding information and financial disclosure

This has been corrected.

4. Financial disclosure

We have corrected this. The Lesotho National Tuberculosis Prevalence Survey was funded by the Global Fund and the World Bank Southern Africa Health System Support project (SATBHSS) through an agreement between the Ministry of Finance, a Global Fund Principal Recipient, and University Research Co., LLC (Contract number: GF/R8/TB/C03-NPBTBPS). The funders had no role in study design, data collection and analysis, decision to publish, or preparation of the manuscript. The survey was implemented by University Research Co. LLC with AQUITY Innovations and Next2People, as subcontractors, who collected and analyzed the data along with the Ministry of Health and WHO. We confirm that there were no competing interests for the authors while preparing this paper. The funding agency did not have any influence in preparing this manuscript. The authors did not receive any remuneration for writing this paper nor were guided by the survey funders to present data or draw conclusions.

We confirm that there were no competing interests for the authors while preparing this paper. The funding agency did not have any influence in preparing this manuscript. The authors did not receive any remuneration for writing this paper nor were guided by the survey funders to present data or draw conclusions.

5. Data availability statement

Data will be publicly available on the Dryad data repository.

6. Images and maps

The map was generated using shapefiles obtained from Natural Earth (public domain): http://www.naturalearthdata.com/

7. Please include your tables as part of your main manuscript and remove the individual files. Please note that supplementary tables (should remain/ be uploaded) as separate "supporting information" files

The tables have now been included in the main manuscript, as requested. 

B. Responses to the Reviewers Comments

1. Is the manuscript technically sound, and do the data support the conclusions?

The authors have revised the manuscript. National TB prevalence surveys have provided critical information that help program managers and policy makers determine if the current disease notification rates determined based on the existing surveillance systems provide an accurate picture of the disease prevalence in the community. This paper presents data from Lesotho on its first national TB prevalence survey. 

2. Is the manuscript presented in an intelligible fashion and written in standard English? Yes

We have revised the document and are confident that we have responded to this issue.

---

## [Editor Report · Decision Letter 1]

10 Jun 2022

PONE-D-22-00269R1Policy and programmatic directions for the Lesotho tuberculosis programme: findings of the national tuberculosis prevalence survey, 2019PLOS ONE

Dear Dr. KAK,

Thank you for submitting your manuscript to PLOS ONE. After careful consideration, we feel that it has merit but does not fully meet PLOS ONE’s publication criteria as it currently stands. Therefore, we invite you to submit a revised version of the manuscript that addresses the points raised during the review process.

Thank you for addressing the reviewers' comments, which you have done well. A minor point still to be improved is use of the word "cases" e.g. in table legends where this should be consistently replaced by "participants with TB" or words to that effect, wherever possible.

We look forward to receiving your revised manuscript.

Kind regards,

Kevin Schwartzman

Academic Editor

PLOS ONE
---

## [Author Response · Author response to Decision Letter 1]

28 Jul 2022

We have replaced "cases" with "survey participants" where appropriate. 

Kind regards,

Neeraj Kak

---

## [Editor Report · Decision Letter 2]

5 Aug 2022

Policy and programmatic directions for the Lesotho tuberculosis programme: findings of the national tuberculosis prevalence survey, 2019

PONE-D-22-00269R2

Dear Dr. KAK,

We’re pleased to inform you that your manuscript has been judged scientifically suitable for publication and will be formally accepted for publication once it meets all outstanding technical requirements.

Kind regards,

Kevin Schwartzman

Academic Editor

PLOS ONE
---

## [Editor Report · Acceptance letter]

28 Feb 2023

PONE-D-22-00269R2 

Policy and programmatic directions for the Lesotho tuberculosis programme: findings of the national tuberculosis prevalence survey, 2019 

Dear Dr. Kak:

I'm pleased to inform you that your manuscript has been deemed suitable for publication in PLOS ONE. Congratulations! Your manuscript is now with our production department. 

Kind regards, 

on behalf of

Dr. Kevin Schwartzman 

Academic Editor

PLOS ONE